Two new sympatric species of Phrynopus (Anura: Strabomantidae) from the Elfin Forests of Cordillera de Yanachaga in central Peru

Venegas Pablo 1 2
García Ayachi Luis Alberto 1 2
Lujan Lesly 2
Duran Vilma 2
Motta Ana motta@ku.edu 3
1 Rainforest Partnership , Austin , TX , United States of America
2 Instituto Peruano de Herpetología (IPH) , Lima , Peru
3 Biodiversity Institute and Natural History Museum, University of Kansas , Lawrence , KS , United States of America
Ereskovsky Alexander
Electronic publication date: 2025 Oct 30
Publication date: 2025
Volume: 13
Electronic Location ID: e20250
Received 2025 Mar 27; Accepted 2025 Sep 25
Copyright: ©2025 Venegas et al.
Copyright year: 2025
Copyright holder: Venegas et al.
License: This is an open access article distributed under the terms of the Creative Commons Attribution License, which permits unrestricted use, distribution, reproduction and adaptation in any medium and for any purpose provided that it is properly attributed. For attribution, the original author(s), title, publication source (PeerJ) and either DOI or URL of the article must be cited.
License URL: https://creativecommons.org/licenses/by/4.0/

Keywords: Andes, Anura, Amphibia, Terrarana, Phylogeny, Singleton species

Funding: Ministerio del Ambiente (MINAM) São Paulo Research Foundation FAPESP; grant #2017/08488-3 The field work was funded by the Ministerio del Ambiente (MINAM). Funds for sequencing were provided by São Paulo Research Foundation (FAPESP; grant #2017/08488-3). The funders had no role in study design, data collection and analysis, decision to publish, or preparation of the manuscript.

==============================
We describe two new sympatric species of the terrestrial-breeding genus Phrynopus (Anura: Strabomantidae) from the elfin forest at 3,280 m a.s.l. in the Cordillera de Yanachaga, Yanachaga-Chemillén National Park, central Peru. Integrating molecular and morphological evidence, we aim to confirm their recognition as new species and assess their generic placement and relationships within Phrynopus. We infer a Maximum-Likelihood phylogeny from five loci (12S, 16S, COI, RAG1, TYR; 4,271 bp of concatenated mtDNA and nuDNA fragments) for 97 terminals, including three representing the new taxa. Phrynopus was recovered as monophyletic, and both new species were placed within a strongly supported subclade that includes P. apumantarum, P. badius, P. barthlenae, P. bracki, P. bufoides, P. horstpauli, P. inti, P. kauneorum, P. miroslawae, P. pesantesi, P. sancristobali, P. tautzorum, and Phrynopus sp. The two new species are not recovered as close sisters but as distinct lineages within this subclade. One of the new species is medium-sized, distinguished by small tubercles on the upper eyelids, tubercles on the heel, a row of tubercles along the outer edge of the tarsus, and red coloration on the groin, thighs, and concealed surfaces of the shanks. The other new species lacks heel and tarsal tubercles and is characterized by its striking black coloration on the groin and hidden surfaces of the hind limbs. Both new species are currently known only from the type locality, where they occur in sympatry with P. miroslawae and P. tribulosus. The discovery of these narrowly distributed species in the Yanachaga-Chemillén National Park, coupled with habitat alteration near the boundaries of the park, highlights the urgent need for effective protection of elfin-forest habitats in the Cordillera de Yanachaga.

Introduction

Terrestrial breeding frogs of the family Strabomantidae are highly diverse, with more than 800 species distributed in tropical and subtropical South America and lower Central America (Frost, 2025). Species of this group deposit their eggs in terrestrial sites where they undergo direct development, lacking the aquatic tadpole stage. This mode of life history, not associated with aquatic environments, is responsible for their success in inhabiting a variety of environments, including cloud forests, lowland forests, dry forest and humid grasslands (Duellman & Lehr, 2009; Frost, 2025; Hedges, Duellman & Heinicke, 2008). Strabomantid species comprise about half of all species of frogs known to inhabit Peru, where they are distributed in 12 genera (Bryophryne, Lynchius, Microkayla, Niceforonia, Noblella, Oreobates, Phrynopus, Phyllonastes, Pristimantis, Qosqophryne, Strabomantis and Yunganastes), occupying a variety of habitats such as Pacific dry forest, humid lowland tropical forests, montane forests, puna and paramo (Duellman & Lehr, 2009; Frost, 2025; Von May et al., 2024).

The high Andean strabomantid frogs share a similar external morphology (the “phrynopoid” morphology) that led them to be historically considered part of a single natural group, the genus Phrynopus (De la Riva, 2020). Phrynopus was re-erected to include 14 species of small frogs, with short limbs and digits without discs, broadly distributed in the high Andes from Colombia to Bolivia (Lynch, 1975). Subsequently, the genus experienced a rapid increase in species described and drastic changes in its composition (De la Riva, 2007; De la Riva, Chaparro & Padial, 2008; Lehr, 2006).

The monophyly of Phrynopus sensu Lynch (1975) was rejected by molecular phylogenetic analyses, revealing a scenario where high-elevation lineages have independent origins (Hedges, Duellman & Heinicke, 2008). The non-monophyletic Phrynopus was split into a number of genera, and the genus Phrynopus was restricted to a clade of 21 species that occur in upper humid forests and grasslands of the Cordillera Oriental in Peru (Hedges, Duellman & Heinicke, 2008). In the following decade, 16 new species of Phrynopus were described and the redefined genus faced another increase in its number of species (Chaparro, Padial & De la Riva, 2008; Chávez et al., 2015; Lehr, Moravec & Cusi, 2012; Lehr & Rodríguez, 2017; Lehr et al., 2017; Mamani & Malqui, 2014; Rodríguez & Catenazzi, 2017; Trueb & Lehr, 2008; Venegas et al., 2018; Von May, Lehr & Rabosky, 2018). With P. curator and P. nicoleae being considered synonyms of P. tribulosus (Von May, Lehr & Rabosky, 2018), P. ayacucho (Chaparro et al., 2007) being transferred to the genus Oreobates (Padial et al., 2012), and the recent description of P. apumantarum, P. remotum, and P. sancristobali, the genus now comprises 37 species, distributed in the Cordillera Oriental and Central of Peru, restricted to a region between 6° and 13° of latitude (Chávez, García-Ayachi & Catenazzi, 2020; Chávez, García-Ayachi & Catenazzi, 2023; Díaz, Mamani & Catenazzi, 2023; Venegas et al., 2018). Species have been recorded at elevations between 2,600–4,490 m, and most of them show very restricted ranges, both horizontally and vertically (Chávez, García-Ayachi & Catenazzi, 2023; Rodríguez & Catenazzi, 2017).

Even with the discovery of numerous new species resulting from recent fieldwork, the diversity of species of Phrynopus is still considered underestimated, as many remote regions of the slopes of the Andes in Peru have not yet been explored and probably are inhabited by additional microendemic taxa (De la Riva et al., 2018). The Yanachaga-Chemillén National Park in the Departamento de Pasco of Peru has a remarkable diversity of amphibians (41 species have been recorded for the area, including undescribed species; Angulo et al., 2016), and fieldwork in the area has led to the discovery of new species of strabomantid frogs, including new species of Phrynopus (Duellman & Hedges, 2005; Duellman & Hedges, 2008; Hedges, 1990; Lehr, Moravec & Cusi, 2012; Lehr et al., 2017). Fieldwork in this region conducted by some of the authors of this study revealed the existence of two unnamed species of Phrynopus. We use phylogenetic analyses of nuclear and mitochondrial genes to assess their phylogenetic relationships and combine morphological and molecular data to support the recognition of the two species described in this study.

Materials & Methods

Morphology, voucher specimens, and permits

Character definition and terminology, and the descriptive scheme of the diagnosis follow that of Duellman & Lehr (2009). Specimens were measured with digital calipers to the nearest 0.1 mm. The following measurements were recorded: snout–vent length (SVL); tibia length (TL); foot length (FL, measured from the proximal edge of the inner metatarsal tubercle to the tip of Toe IV); head length (HL, taken obliquely from the jaw angle to the snout tip); head width (HW, at the jaw angle); eye diameter (ED); interorbital distance (IOD); upper eyelid width (EW); internarial distance (IND); and eye–nostril distance (E–N, the straight-line distance between the anterior corner of the orbit and the posterior margin of the external naris). Fingers are numbered from I–IV, starting preaxially. To compare relative lengths, Fingers I and II were pressed together, while Toes III and V were each placed against Toe IV. Specimens were preserved in 10% formalin and stored in 70% ethanol. Tissue samples were collected before fixing specimens with formalin and preserved in 95% ethanol. Specimens were sexed externally by the presence or absence of vocal slits and internally by the condition of the gonads. All specimens were deposited in the herpetological collection of the Centro de Ornitología y Biodiversidad (CORBIDI), Lima, Peru. Institutional abbreviations follow Sabaj (2020). We obtained our research permit through the Dirección General Forestal y de Fauna Silvestre, Ministerio de Agricultura y Riego, Peru, which issued the Contrato de Acceso Marco a Recursos Genéticos, numbered 359-2013-MINAGRI-DGFFS-DGEFFS. Our research was approved by the Institutional Animal Care and Use Committee of University of Kansas (AUS 279-01). Specimens examined are listed in Appendix S1.

DNA extraction and sequencing

We extracted total DNA from ethanol preserved tissues, following standard high-salt protocol adapted for microcentrifuge tubes (Lyra, Haddad & De Azeredo Espin, 2017; Maniatis, Fritsch & Sambrook, 1982). We amplified two mitochondrially encoded gene fragments: one including the partial sequences of 12S rRNA, tRNA-val and 16S rRNA genes (H1 fragment) and a fragment of the cytochrome c oxidase I (COI); and two nuclear genes: partial sequences of tyrosinase (TYR), and partial sequences of recombination activating 1 (RAG1). Amplifications were carried out in a 22 µl reaction using Ampliqon Taq DNA Polymerase Master Mix (Ampliqon A/S, Odense M, Denmark), with primers listed in Table S1. For the mitochondrial genes we followed polymerase chain reactions (PCR) conditions described in Lyra, Haddad & De Azeredo Espin (2017) (set-up reaction (UP reaction) protocol). For nuclear genes we used the following PCR cycling protocol: 3 min of denaturation at 95 °C, followed by 45 cycles of 20 s of denaturation at 95 °C plus 20s of annealing at 56 °C plus 1 min of extension at 68 °C, followed by 3 min of final extension at 68 °C, and stored at 12 °C. PCR products were purified using an enzymatic reaction containing 1 unit of Exonuclease I and 0.5 unit of Alcaline phosphatase (Thermo Fisher Scientific Inc.) and were sent to Macrogen, Inc., Seoul, Repuiblic of Korea, for sequencing. Sequence files were checked for quality and contigs were assembled using Geneious R11 (Biomatters).

Phylogenetic analysis

We used phylogenetic trees to assess generic assignment and investigate the relationship of the focal samples. We supplemented our sequences with sequences of the mitochondrial 12S rRNA and partial sequence of 16S rRNA genes, and the protein-coding gene cytochrome c oxidase subunit I (COI) as well as nuclear genes recombination-activating gene 1 (RAG1) and tyrosinase precursor (TYR) available on GenBank belonging to species of Phrynopus. One new species is represented by the terminal CORBIDI 7379, and the other one by the terminals CORBIDI 7382 and CORBIDI 7385. Our ingroup sample includes 63 terminals of Phrynopus representing 24 nominal species, two unnamed species (Phrynopus spI and Phrynopus sp. of Von May, Lehr & Rabosky, 2018), and the two species we name herein. As outgroups we included one terminal per species of the related genera Lynchius (n = 8) and Oreobates (n = 25) and rooted all our analyses with the distantly related species Haddadus binotatus (Padial, Grant & Frost, 2014). Specimen voucher numbers for newly produced sequences and GenBank accession numbers for all sequences used in this study are listed in Appendix S2.

We performed multiple sequence alignments in MAFFT online v7 using the global instreaming iterative (G-INS-i) strategy, which is considered appropriate for alignments that consist of large numbers of sequences (Katoh & Standley, 2013).

We generated Maximum Likelihood (ML) phylograms using IQ-TREE v1.6 (Nguyen et al., 2015) from the concatenated sequence of the five gene fragments included in our dataset. We determined the best-fit substitution model for each gene via ModelFinder, implemented within IQ-TREE (Kalyaanamoorthy et al., 2017) and performed a partitioned analysis according to codon position within the protein-coding genes (COI, RAG1, and TYR). We calculated branch support with 10,000 bootstrap replicates using the Ultrafast Bootstrapping algorithm (Hoang et al., 2018). Alignments, script, and output files, including partitions and trees, are available as supplementary files.

Nomenclatural act

The electronic version of this article in Portable Document Format (PDF) will represent a published work according to the International Commission on Zoological Nomenclature (ICZN), and hence the new names contained in the electronic version are effectively published under that Code from the electronic edition alone. This published work and the nomenclatural acts it contains have been registered in ZooBank, the online registration system for the ICZN. The ZooBank LSIDs (Life Science Identifiers) can be resolved and the associated information viewed through any standard web browser by appending the LSID to the prefix http://zoobank.org/. The LSID for this publication is: urn:lsid:zoobank.org:pub:513EDEF4-3F07-4097-8902-8DFED468E4C3. The online version of this work is archived and available from the following digital repositories: PeerJ, PubMed Central SCIE and CLOCKSS.

Results

Phylogenetic relationships

The optimal similarity-alignment of our concatenated dataset comprises 4,271 character columns for 97 terminals. Our analysis recovered the genera Phrynopus as monophyletic, with 100% bootstrap support and supported the placement of the two new species in the genus Phrynopus (Fig. 1). The two new species are recovered as part of a highly supported clade (90%) that includes P. apumantarum, P. badius, P. barthlenae, P. bracki, P. bufoides, P. horstpauli, P. inti, P. kauneorum, P. miroslawae, P. pesantesi, P. sancristobali, P. tautzorum and Phrynopus sp. The phylogenetic position and morphological distinctiveness of the newly collected specimens support the description of the two new species, which we name and diagnose below.

Figure 1 Phylogenetic tree resulting from the analysis of data sets of 4,271 aligned bp and composed of the mitochondrial genes 12S, 16S, and COI, and fragments of the nuclear protein-coding genes RAG1 and TYR.

Maximum likelihood optimal tree (log likelihood −35,013.622762) and bootstrap node values. Asterisks represent values of 100.

Phrynopus manuelriosi sp. nov.	
urn:lsid:zoobank.org:act:86453AE1-9A6A-4ABD-9681-CF9E372B877D	
Figs. 2–4; Table 1	

Holotype

CORBIDI 7385 (Fig. 2), adult female, Santa Bárbara, Distrito de Huancabamba, Provincia de Oxapampa, Departamento de Pasco, Peru, (10°20′29.1″S, 75°38′27.1″W, 3,280 m a.s.l.), collected by Pablo J. Venegas, Vilma Duran, Caroll Z. Landauro, and Lesly Lujan, on 25 August 2010.

Figure 2 Holotype of Phrynopus manuelriosi sp. nov. (CORBIDI 7385; SVL 26.1 mm) in life.

(A) Dorsolateral, and (B) ventral views.

Figure 3 Preserved holotype of Phrynopus manuelriosi sp. nov.

(A) Dorsal view, (B) ventral view, (C) palm, (D) sole, and (E) head in lateral view. Scale five mm. Photographs by LAGA.

Figure 4 Paratypes of Phrynopus manuelriosi sp. nov. in life.

(A) Dorsolateral (B) lateral (showing the groin and anterior surface of thigh), and (C) ventral views of CORBIDI 7380 (SVL 25.6 mm); (D) dorsolateral, (E) dorsal (showing a fine pale middorsal band, red groin, and the red anterior and posterior surface of thigh), and (F) ventral views of CORBIDI 7383 (SVL 27.1 mm); (G) dorsolateral and (H) ventral views of male paratype CORBIDI 7390 (SVL 13.2 mm); (I) dorsolateral and (J) ventral views of male paratype CORBIDI 7386 (SVL 17.4 mm); (K) dorsolateral view of female CORBIDI 7387 (SVL 19.4 mm); (L) dorsolateral view of male CORBIDI 7381 (SVL 16.1 mm).

Paratypes (9)

Nine specimens in total, four adult females (CORBIDI 7380, 7382–83, 7387) and five adult males (CORBIDI 7381, 7386, 7388–90), same data as the holotype.

Diagnosis

A species in the genus Phrynopus characterized by (1) skin on dorsum shagreen with scattered low tubercles, more abundant and prominent on flanks and hind limbs; usually bearing interorbital fold, \/-shaped fold on scapular region, and /\-shaped fold on the middle of dorsum; skin on venter areolate; groins smooth or weakly areolate; dorsolateral folds absent; supratympanic fold conspicuous and long, slightly curved above the tympanic region; discoidal fold present only as thoracic fold or completely absent; (2) tympanic membrane and annulus absent; (3) snout moderately short, bluntly rounded in dorsal view and in profile; (4) upper eyelid bearing small tubercles, narrower than IOD; cranial crests absent; (5) vomerine teeth absent; (6) vocal slits present and nuptial pads absent; vocal sac absent; (7) Finger I shorter than Finger II; tips of fingers rounded and narrow; (8) fingers lacking lateral fringes; subarticular tubercles small and rounded in dorsal view, and flat on lateral view; (9) ulnar tubercle present, more evident in males; (10) heel bearing one or two subconical tubercles and outer edge of tarsus bearing a row of broad conical tubercles; inner tarsal fold absent; (11) inner metatarsal tubercle ovoid, about equal in size to rounded outer metatarsal tubercle; subarticular tubercles small and rounded in dorsal view, flat on lateral view; supernumerary plantar tubercles present; (12) toes lacking lateral fringes; webbing absent; Toe V longer than Toe III; tips of toes rounded; (13) in life, dorsum of head, body, and limbs pale brown, yellowish-brown or grayish-brown with dark brown markings; groin, anterior and posterior surface of thighs, and concealed surface of shanks red; ventral surface yellowish-brown, grayish-brown or yellow with or without dark brown flecks on the throat and belly; iris red with black reticulations; (14) SVL in five males 11.3–17.4 mm, in five females 19.4–27.1 mm.

Comparisons

Among the 37 described species of Phrynopus, only P. badius, P. bracki, P. daemon, P. heimorum, P. inti, P. paucari, P. peruanus, P. unchog and P. vestigiatus show reddish coloration in the groins (Duellman & Lehr, 2009; Lehr, 2001; Lehr, Moravec & Cusi, 2012; Lehr & Oroz, 2012; Lehr & Rodríguez, 2017). Phrynopus manuelriosi sp. nov. differs from P. badius, P. bracki, P. inti, P. paucari and P. vestigiatus by having uniformly red groin (groin dark brown with bright orange flecks in P. badius; brown with red spots in P. bracki; pale grayish with salmon-colored flecks in P. inti; greenish yellow with diffuse salmon blotches in P. paucari; and dark brown with red well-defined blotches in P. vestigiatus). Of the species sharing a uniformly reddish coloration in the groin, P. manuelriosi sp. nov. can be distinguished by the presence of heel tubercle (absent in P. daemon, P. heimorum, and P. peruanus), tarsal tubercles (absent in P. heimorum and P. peruanus), and eyelid tubercles (absent in P. daemon, P. heimorum, P. peruanus, and P. unchog). Moreover, P. manuelriosi sp. nov. lacks tympanic membrane and annulus, which are present in P. peruanus.

The presence of tubercles on the heel and outer edge of tarsus is uncommon in the genus Phrynopus. Only six species (P. bracki, P. dagmarae, P. kotosh, P. oblivious, P. tribulosus and P. vestigiatus) share the presence of tubercles on the heel and outer edge of tarsus. Phrynopus manuelriosi sp. nov. differs from P. bracki and P. dagmarae by having tubercles on the upper eyelids (absent in P. bracki and P dagmarae), and P. dagmarae has the Toe V shorter than Toe III, while the Toe V is larger than Toe III in the new species. Phrynopus dagmarae, P. kotosh and P. oblivius also differ by lacking \/-shaped fold in the scapular region and a /\-shaped fold on the middle of the dorsum. Moreover, P. manuelriosi sp. nov. lacks dorsolateral folds, whereas dorsolateral folds are present in P. dagmarae (continuous), P. kotosh (discontinuous), and P. vestigiatus (prominent and undulated). Phrynopus tribulosus has Toe V equal or slightly shorter than Toe III (Von May, Lehr & Rabosky, 2018), while in P. manuelriosi sp. nov. the Toe V is longer than Toe III. In addition, P. barthlenae and P. miroslawae have tubercles on the heel but lack tubercles on the outer edge of tarsus (present in P. manuelriosi sp. nov.) and Toe V is shorter than Toe III in P. barthlenae and equal in length in P. miroslawae, respectively, while P. manuelriosi sp. nov. has Toe V longer than Toe III.

Phrynopus horstpauli has a habitus similar to P. manuelriosi sp. nov., also being found in leaf and branches of the understory (Duellman & Lehr, 2009; Lehr, Köhler & Ponce, 2000) and sharing the slender limbs and relative long narrow fingers and toes. Phrynopus manuelriosi sp. nov. can be easily distinguished from P. horstpauli by having one or two subconical tubercle on the heel and a row of conical tubercles on the outer edge of the tarsus (absent in P. horstpauli), and a smaller size with a SVL of 11.3 to 17.4 mm in males and 19.4 to 27.1 mm in females (17.7 to 25.6 mm in males and 30.8 to 39.7 mm in females of P. horstpauli).

Phrynopus melanoinguinis sp. nov., described bellow, occurs in sympatry with P. manuelriosi sp. nov. and it can be easily distinguished by lacking heel and tarsal tubercles (present in P. manuelriosi sp. nov.), and by having dorsolateral fold (absent in P. manuelriosi sp. nov.).

Due to the variable coloration of P. manuelriosi sp. nov., we consider it possible to confuse it with Noblella duellmani, a geographically close species. Noblella duellmani occurs in Departamento de Pasco, at elevations between 2,900 and 3,500 m, the same elevational range of P. manuelriosi sp. nov. Noblella duellmani can be easily distinguished from P. manuelriosi sp. nov. by having Toe V shorter than Toe III (Toe V longer than Toe III in P. manuelriosi sp. nov.), tips of digit slightly expanded and those of Toes III–V slightly acuminate (tips of toes narrow and rounded in P. manuelriosi sp. nov.), and skin of belly smooth (areolate in P. manuelriosi sp. nov.).

Table 1 Variation of measurements (in mm) and proportions of the type series of Phrynopus manuelriosisp. nov.

Abbreviations are as follow: SVL, snout–vent length; TL, tibia length; FL, foot length; HL, head length; HW, head width; ED, eye diameter; IOD, interorbital distance; EW, upper eyelid width; IND, internarial distance, and E-N, eye–nostril distance.

Measurements and proportions	Females N = 5	Males N = 5	
SVL	19.4–27.1 (24.8 ± 3.1)	11.3–17.4 (14.3 ± 2.4)	
TL	9.7–11.9 (11.2 ± 0.8)	6–8.7 (7.2 ± 1.1)	
FL	9.9–14.1 (12.2 ± 1.5)	5.9–9.5 (7.4 ± 1.5)	
HL	7.8–10.2 (9.2 ± 0.9)	4.4–7.2 (5.7 ± 1.2)	
HW	7.6–9.3 (8.7 ± 0.6)	3.9–6.6 (5.3 ± 1.7)	
ED	2.7–3.3 (2.9 ± 0.3)	1.7–2.3 (2.2 ± 0.2)	
IOD	2.7–3.1 (2.9 ± 0.1)	1.8–2.5 (2.6 ± 0.2)	
EW	1.7–2.6 (2.1 ± 0.3)	1.4–1.9 (1.6 ± 0.1)	
IND	1.5–1.9 (1.7 ± 0.1)	1.0–1.7 (1.2 ± 0.2)	
E–N	1.6–1.9 (1.7 ± 0.1)	1.0–1.3 (1.1 ± 0.1)	
TL/SVL	0.4–0.5 (0.4 ± 0.3)	0.4–0.5 (0.5 ± 0.2)	
FL/SVL	0.4–0.5 (0.4 ± 0.4)	0.4–0.5 (0.5 ± 0.4)	
HL/SVL	0.3–0.4 (0.3 ± 0.2)	0.3–0.4 (0.6 ± 0.2)	
HW/SVL	0.3–0.3 (0.3 ± 0.3)	0.3–0.3 (0.8 ± 0.1)	
HW/HL	0.8–1.4 (0.9 ± 0.7)	0.8–1.2 (0.8 ± 0.5)	
E–N/ED	0.5–0.7 (0.5 ± 0.7)	0.4–0.7 (0.4 ± 0.1)	
EW/IOD	0.6–0.8 (0.7 ± 0.9)	0.6–0.9 (0.8 ± 0.1)	

Description of holotype

Adult female (Figs. 2 and 3); body moderately robust; head about as wide as body, nearly as long as wide; snout bluntly rounded in dorsal view and in profile; canthus rostralis slightly curved in dorsal view, rounded in profile; loreal region nearly flat; lips rounded; nostrils barely protuberant, directed laterally; internarial region barely depressed; top of head flat; width of upper eyelid narrower than interorbital distance (EW/IOD 0.84); eye large, its diameter much greater than its distance from nostril (E-N/ED 0.70); tympanic membrane and annulus absent; supratympanic fold distinct, angling posteroventrally from point behind the tympanic region close to the arm insertion; postrictal tubercles present, rounded. Tongue longer than broad, notched posteriorly, posterior half free; choanae small, round, not concealed by palatal shelf of maxillary; dentigerous processes of vomers absent.

Forelimb slender; ulnar tubercles low, diffuse; palmar tubercle low, round, about same size as thenar tubercle; subarticular tubercles distinct, small and rounded in dorsal view, and flat in lateral view; supernumerary tubercles present, weakly defined; fingers slender and long, lacking lateral fringes; relative lengths of fingers I < II < IV < III; tips of fingers narrow, rounded, lacking circumferential grooves. Hind limb long and slender; heel bearing two small subconical tubercles and tarsus bearing three low conical tubercles, broad at their base; inner tarsal fold present on distal half of tarsus; inner metatarsal tubercle elevated, round, about twice the size of subconical outer metatarsal tubercle; subarticular tubercles distinct only at the base of toes, small and rounded in dorsal view, flat in lateral view; supernumerary tubercles present, distinct; toes slender, lacking lateral fringes; relative lengths of toes I < II < III < V < IV; tips of toes narrow, rounded, lacking circumferential grooves. Skin on dorsum shagreen with scattered low tubercles posteriorly, bearing an interorbital fold, a V-shaped fold on scapular region, and a / \-shaped fold on the sacrum; flanks and hind limbs tuberculate; upper eyelids bearing low rounded tubercles; skin on venter areolate; thoracic fold present; skin ventral and ventrolateral to cloaca granular.

Measurements (in mm) and proportions of holotype: SVL 26.1; TL 11.5; HW 9.3; HL 8.9; IOD 3.1; IND 1.8; EW 2.6; FL 14.1; ED 2.7; E-N 1.9; TL/SVL 0.44; FL/SVL 0.54; HL/SVL 0.34;

In life, dorsal coloration of head, body, and limbs pale brown with pale and dark markings that include a dark brown interorbital bar with a cream border in the anterior margin, a brownish-cream V-shaped fold with a dark brown border in the posterior margin, one dark brown chevron with brownish-cream borders in the middle of the dorsum, one diagonal stripe on the flanks with a pale cream border, and two transverse bars on the hind limbs; dark brown head markings include a bold canthal and supratympanic stripes with cream borders, and a bold labial bar below the eyes with a cream border in the posterior margin (Fig. 2A); groin, anterior and posterior surface of thighs, and concealed surface of shanks red; ventral surface yellowish-brown with palms, soles and ventral surface of thighs brown (Fig. 2B); iris dark or light bronze with fine black reticulations and a faint reddish stripe across the middle.

In preservative, dorsum of head, body, limbs, and sides of head grayish-brown with the same dark brown markings and the same pale borders (now grayish-cream) such as described above; groin, anterior and posterior surface of thighs, and concealed surface of shanks grayish-brown; ventral surface on throat and chest pale tan, belly grayish-cream, with palm, soles, and limbs brown (Fig. 3).

Variation

Sexual dimorphism is evident in respect to snout-vent length, with males smaller than females: SVL 11.3–17.4 mm in males and 19.4–27.1 mm in females (Table 1). In life, the ventral coloration of males and females can vary from brownish-cream to yellowish-cream (Figs. 4C, 4F, and 4J); only one male (CORBIDI 7381) has a grayish venter with dark brown flecks in the throat and belly (Fig. 4H). The dorsal coloration is variable: one female (CORBIDI 7380) has a dull brown dorsum without distinct markings except for a dark brown interorbital bar (Fig. 4A); one female (CORBIDI 7383) has a narrow pale middorsal stripe (Fig. 4E); one male (CORBIDI 7381) has a grayish-brown dorsum with dark brown blotches, a black labial bar below the eyes and lack canthal stripe (Fig. 4G); one male (CORBIDI 7381) has dark yellow flanks (Fig. 4L); one female (CORBIDI 7387) has an orange hue on dorsum with the markings similar to those of the holotype except for the canthal stripe (Fig. 4K); one male (CORBIDI 7386) has a distinct black canthal and supratympanic stripe (Fig. 4I).

Distribution and natural history

Phrynopus manuelriosi sp. nov. is only known from the type locality in the west margin of Río Huancabamba at an elevation of 3,280 m a.s.l., Provincia de Oxapampa, Departamento de Pasco, on the eastern slope of Cordillera Oriental in central Peru (Fig. 5). Eighteen individuals, of which ten were collected, were found in 5 hours surveying amphibians at night by four collectors. All individuals were found on the ground in the elfin forest, and perched on leaves and branches about 20–100 cm above the ground in the forest and in the riparian vegetation. The herpetological survey in Santa Bárbara occurred in the dry season and no rainfall was recorded during the four days of survey in this locality. The two new species described herein were found in sympatry in the elfin forest of Santa Bárbara. Gastrotheca griswoldi also occurs in the area, but above tree line in the Puna grasslands. Two other species of Phrynopus are known to occur in the same area of Santa Bárbara: P. miroslawae and P. tribulosus, although we did not observe them during our surveys. Phrynopus miroslawae is found in the elfin forest and might be syntopic with P. manuelriosi sp. nov., whereas P. tribulosus inhabits the Puna grasslands.

Figure 5 Map of the National Park Yanachaga Chemillen showing the distribution of all species of Phrynopus that inhabit within the park limits.

The type locality of Phrynopus manuelriosi sp. nov. new species and P. melanoinguinis sp. nov. is represented by a star.

Etymology

The name is a patronym for Manuel Ríos, a Peruvian forest engineer and professor at the Faculty of Forestry at Universidad Nacional Agraria La Molina (UNALM), Lima, Peru, from 1970 to 2017, who has dedicated his life to preserving the natural heritage of his country. As professor, Manuel trained hundreds of students, inspiring them to become committed advocates for resource conservation, wildlife management and the protection of natural areas. He is also a founder and life member of the Board of Directors of the Peruvian Foundation for the Conservation of Nature (Pro Naturaleza), an organization that has played a key role in the preservation and protection of the environment in Peru. Likewise, he was Director of the Conservation Data Center (CDC-UNALM) between 1983 and 1998, and his legacy is present in the creation and planning of some of the most emblematic protected areas in Peru: the Paracas National Reserve, the Titicaca National Reserve, the Lachay National Reserve, the Abiseo National Park, and the Tabaconas Namballe National Sanctuary, among many others.

Phrynopus melanoinguinis sp. nov. urn:lsid:zoobank.org:act:13A87303-F889-41BD-A67E-F00D4C611F17	
Figs. 6–7	

Holotype

CORBIDI 7379 (Fig. 6), adult female, Santa Bárbara, Distrito de Huancabamba, Provincia de Oxapampa, Departamento de Pasco, Peru (10°20′29.1″S, 75°38′27.1″W, 3,280 m a.s.l.), collected by Pablo J. Venegas on 25 August 2010.

Figure 6 Holotype of Phrynopus melanoinguinis. sp. nov. (CORBIDI 7379) in life.

(A) Dorsolateral, (B) lateral (showing the characteristic black coloration on the groin), and (C) ventral views (SVL 23.6 mm).

Figure 7 Preserved holotype of Phrynopus melanoinguinis sp. nov.

(A) Dorsal view, (B) ventral view, (C) palm, (D) sole, and (E) head in lateral view. Scale five mm. Photographs by LAGA.

Diagnosis

A species in the genus Phrynopus characterized by (1) skin on dorsum smooth with scattered granules; flanks and venter areolate; dorsolateral folds present, short; discoidal fold present only as thoracic fold; supratympanic fold conspicuous and long, slightly curved above the tympanic region; (2) tympanic membrane and annulus absent; (3) snout moderately short, bluntly rounded in dorsal view and in profile; (4) upper eyelids narrower than IOD, bearing low small tubercles; cranial crests absent; (5) vomerine teeth absent; (6) males unknown; (7) Finger I shorter than Finger II; tips of fingers rounded and narrow; (8) fingers lacking lateral fringes; subarticular tubercles small, rounded, weakly defined in dorsal view and flat in lateral view; supernumerary tubercles present, weakly defined; (9) ulnar tubercles absent; (10) heel and outer edge of tarsus lacking tubercles; inner tarsal fold absent; (11) inner metatarsal tubercle ovoid, prominent, about equal in size to lower, rounded, outer metatarsal tubercle; subarticular tubercles small, round, distinct only at the base of toes in dorsal view and flat on lateral view; supernumerary plantar tubercles absent; (12) toes lacking lateral fringes; webbing absent; Toe V slightly longer than Toe III; tips of toes rounded; (13) in life, dorsum dark brown without marks; groin, anterior and posterior surface of thighs, and concealed surface of shanks black; ventral surface brown with a black blotch on the throat; iris bluish-gray with fine black reticulation; (14) SVL of single female 23.6 mm.

Comparisons

Phrynopus melanoinguinis sp. nov. is strikingly different from any other known species in the genus by having the groin and the hidden surfaces of the hind limbs black, and blueish-gray iris in life. Phrynopus melanoinguinis sp. nov. occurs in sympatry with P. manuelriosi sp. nov., P. miroslawae, and P. tribulosus, and it differs from those by lacking heel and tarsal tubercles, and by having dorsolateral and supratympanic folds (absent in P. manuelriosi sp. nov. and P. tribulosus). Three other species of Phrynopus occur within the Yanachaga-Chemillén National Park: P. auriculatus, P. badius, and P. bracki (Chaparro, Padial & De la Riva, 2008; Duellman & Lehr, 2009; Lehr & Oroz, 2012). P. melanoinguinis sp. nov. differs from those species by lacking heel and tarsal tubercles (heel tubercle present in P. auriculatus and P. bracki; tarsal tubercle present in P. bracki), by having eyelid tubercles (absent in P. auriculatus, P. badius, and P. bracki), and by having dorsolateral folds (absent in P. bracki) and supratympanic folds (absent in P. badius, and P. bracki).

The absence of heel and tarsal tubercles also distinguishes P. melanoinguinis sp. nov. from P. daemon, P. dagmarae, P. interstinctus, P. vestigiatus, P. kotosh, P. oblivius, P. remotum, and P. unchog. The combination of dorsolateral and long supratympanic folds also differentiates P. melanoinguinis sp. nov. from many other species in the genus: P. barthlenae, P. bufoides, P. capitalis, P. chaparroi, P. daemon, P. dagmarae, P. heimorum, P. interstinctus, P. inti, P. juninensis, P. kauneorum, P. lapidoides, P. lechriorhyncus, P. montium, P. oblivious, P. peruanus, P. pesantesi, P. remotum, P. sancristobali, P. tautzorum, P. thompsoni, P. valquii, and P. vestigiatus. In the case of P. bufoides and P. sancristobali, both species are also easily distinguished from P. melanoinguinis sp. nov. by the presence of conspicuous large round or elongate pustules on dorsum and flanks (dorsum smooth with areolate flanks in the new species). Moreover, the absence of tympanic membrane and annulus distinguishes the new species from P. auriculatus, P. peruanus and P. mariellaleo (tympanic membrane and annulus present).

Phrynopus paucari differs from P. melanoinguinis sp. nov. by having larger subconical tubercles forming discontinuous longitudinal ridges dorsolaterally (dorsal skin smooth) and venter greenish yellow with brown reticulation (dull brown). Phrynopus pesantesi differs from P. melanoinguinis sp. nov. by having ulnar tubercles (absent) and the venter brown with gray mottling (dull brown). Phrynopus horstpauli differs from P. melanoinguinis sp. nov. by having the skin of dorsum slightly tuberculate (smooth), Toe V much longer than Toe III (Toe V slightly longer than Toe III), and venter cream with brown blotches (dull brown). Phrynopus barthlenae, P. heimorum and P. tautzorum differ from P. melanoinguinis sp. nov. by having Toe III larger than Toe V, while Toe III is shorter than Toe V in P. melanoinguinis sp. nov. Furthermore, the dorsum is coarsely tuberculate in P. barthlenae and P. apumantarum, and smooth in P. melanoinguinis sp. nov. In P. apumantarum, the skin on the entire venter is coarsely areolate, while in P. melanoinguinis sp. nov. is areolate peripherally and smooth in the center. Phrynopus kauneorum and P. lechriorhynchus differ from P. melanoinguinis sp. nov. by the presence of dentigerous processes of vomers, whereas these are absent in P. melanoinguinis sp. nov.; in addition, P. kauneorum lacks dorsolateral fold (dorsolateral fold present in the new species), while P. lechriorhynchus has the snout spatulate, long and depressed, broadly rounded in dorsal view and sloping anteroventrally in profile (snout short and bluntly rounded in dorsal view and in profile in P. melanoinguinis sp. nov.).

In addition, P. personatus from the Río Abiseo National Park (Departamento San Martín) in northern Peru is similar to P. melanoinguinis sp. nov. in that both species have the groins and hidden surfaces of hind limbs black ( Rodríguez & Catenazzi, 2017). However, P. personatus has the black surfaces of groins and hind limbs adorned by conspicuous white blotches and the skin of dorsum shagreen with scattered tubercles (dorsum smooth in P. melanoinguinis sp. nov.).

Description of the holotype

Adult female (Figs. 6 and 7); body moderately robust; head narrower than body, nearly as long as wide; snout bluntly rounded in dorsal view and in profile; canthus rostralis slightly curved in dorsal view, rounded in profile; loreal region nearly flat; lips rounded; nostrils barely protuberant, directed laterally; internarial region flat; top of head flat; width of upper eyelid narrower than IOD (EW/IOD 0.74); eye large, its diameter greater than its distance from nostril (E-N/ED 0.63); tympanic membrane and annulus, absent; supratympanic fold conspicuous and long, slightly curved above the tympanic region; ovoid postrictal tubercles present, minute. Tongue slightly longer than broad, not notched posteriorly, posterior half free; choanae small, rounded, not concealed by palatal shelf of maxillary; dentigerous processes of vomers absent.

Forelimb slender; ulnar tubercles absent; palmar tubercle low, round, slightly longer than thenar tubercle; subarticular tubercles small and rounded in dorsal view, and flat on lateral view; two supernumerary tubercles present, weakly defined; fingers short and slender, lacking lateral fringes; relative lengths of fingers I < II < IV < III; tips of fingers narrow, rounded, lacking circumferential grooves. Hind limb slender; heel and tarsus lacking tubercles; inner tarsal fold absent; inner metatarsal tubercle prominent, round, about twice as much of round outer metatarsal tubercle; subarticular tubercles small, rounded, weakly defined in dorsal view and flat in lateral view; supernumerary tubercles absent; toes slender, lacking lateral fringes; relative lengths of toes I < II < III < V < IV; tips of toes narrow, rounded, lacking circumferential grooves. Skin on dorsum smooth with scattered low and round tubercles, flanks areolate, and hind limbs tuberculate; dorsolateral fold present and short; upper eyelids bearing low small tubercles; skin of throat and chest areolate, on belly weakly areolate in the center and coarsely areolate peripherally; discoidal fold absent, thoracic fold present; skin ventral and ventrolateral to cloaca granular.

Measurements (in mm) and proportions of holotype: SVL 23.6; TL 8.9; HW 8.7; HL 9.1; IOD 2.7; IND 1.8; EW 2; FL 9.3; ED 2.7; E-N 1.7; TL/SVL 0.38; FL/SVL 0.39; HL/SVL 0.39; HW/SVL 0.37; HW/HL 0.96; E-N/ED 0.63; EW/IOD 0.75.

In life, dorsal surface of head, body and limbs, and flanks reddish-brown (Fig. 6A); anterior and posterior surface of thighs, groin, and concealed surface of tibia black (Fig. 6B); ventral surface pale reddish-brown, with a dark brown blotch on the throat (Fig. 6C); iris bluish-gray with fine black reticulation.

In preservative, dorsum dark brown without marks; groin, anterior and posterior surface of thighs and concealed surface of shanks black; ventral surface brown with a black blotch on the throat (Fig. 7).

Distribution and natural history

Phrynopus melanoinguinis sp. nov. is known only from the type locality and is syntopic with P. manuelriosi sp. nov. The type locality for both new species described here is within the Peruvian Yanachaga-Chemillén National Park (Fig. 5). The single specimen collected was found on a mossy ground at night in the elfin forest habitat. Phrynopus melanoinguinis sp. nov. might be syntopic with P. miroslawae and sympatric with P. tribulosus.

Etymology

The specific name is an adjective derived from the Greek melano (meaning black) and the Latin inguinis (meaning groin) and is used as a noun in apposition. The name refers to the species’ distinctive black groin.

Discussion

Most species of Phrynopus have a cryptic mode of life restricted to leaf litter and moss layers, making it difficult to find and observe individuals in the field (Lehr & Oroz, 2012). Consequently, many species are known from a limited number of specimens, reflecting the rarity of some species in this genus (Lehr & Oroz, 2012; Rodríguez & Catenazzi, 2017), which leads to descriptions based on a few or even a single specimen, including our description of P. melanoinguinis sp. nov. The challenge of describing a species based on few specimens is related to the lack of understanding of the intraspecific variation in diagnostic characters (Köhler & Padial, 2016). However, that can be mitigated by using multiple lines of evidence (e.g., morphological and molecular in the case of P. melanoinguinis sp. nov.) to name well-supported singleton species.

While Phrynopus melanoinguinis sp. nov. is known only from the female holotype, the type series of P. manuelriosi sp. nov. is comparatively large (five females and five males). Limited number of male specimens seems to be recurrent in Phrynopus, as for three other species only females are known (P. miroslawae, P. thompsoni and P. vestigiatus), while in eight species only one male has been collected (P. lapidoides, P. unchog, P. anancites, P. capitalis, P. personatus, P. daemon and P. chaparroi). This broader pattern across the genus could represent actual female-to-male ratios in populations, but could also be related to natural history aspects of the species. Even though little is known about the reproductive behavior of most species of Phrynopus, the “secretiveness of males” can explain why males are harder to find (Lehr, 2001). For example, males of P. bracki have been reported to call from hidden places in leaf litter and moss vegetation (Hedges, 1990) and males of P. peruanus were found calling from inside grass tussocks (Chaparro et al., 2007). Moreover, males of P. badius and P. tribulosus were heard calling, but could not be located in the dense vegetation, despite the collectors’ efforts (Lehr, Moravec & Cusi, 2012).

The occurrence of two or more species of Phrynopus in the same locality was considered rare, but recent surveys have reported many cases where species of Phrynopus co-occur in the same region (see Chaparro, Padial & De la Riva, 2008; Chávez et al., 2015; Duellman & Hedges, 2008; Lehr, 2001; Lehr & Aguilar, 2002; Lehr, Aguilar & Köhler, 2002; Lehr, Fritzsch & Müller, 2005; Lehr, Moravec & Cusi, 2012; Lehr & Oroz, 2012; Lehr & Rodríguez, 2017; Rodríguez & Catenazzi, 2017; Von May, 2017). In many cases, species segregate by elevation or by microhabitat type (Rodríguez & Catenazzi, 2017). Phrynopus manuelriosi sp. nov. and P. melanoinguinis sp. nov. were found in sympatry in the elfin forest at 3,280 m elevation, but apparently show some microhabitat segregation, since only P. manuelriosi sp. nov. is known to perch on leaves and branches about 20–100 cm above the ground in the forest and in the riparian vegetation.

Figure 8 Map of Peru showing the distribution of species in the genus Phrynopus.

1. P. mariellaleo; 2. P. thompsoni; 3.P. capitalis; 4.P. dumicola; 5.P. personatus; 6. P. anancites; 7. P. valquii; 8. P. remotum; 9.P. daemon; 10. P. lapidoides; 11. P. unchog; 12. P. vestigiatus; 13. P. lechriorhynchus; 14. P. kauneorum; 15. P. dagmarae; 16. P. interstinctus; 17. P. horstpauli; 18. P. heimorum; 19.P. miroslawae; 20. P. tautzorum; 21. P. barthlenae; 22. P. badius; 23. P. tribulosus; 24. P. pesantesi; 25.P. auriculatus; 26. P. bracki ; 27.P. paucari; 28. P. juninensis; 29. P. bufoides; 30. P. kotosh; 31. P. montium; 32. P. peruanus ; 33. P. oblivius; 34. P. inti; 35.P. chaparroi. 36. P. apumantarum. 37. P. sancristobali. The green star represents the type locality of P. manuelriosi sp. nov. and P. melanoinguinis sp. nov.

These findings support previous hypotheses about high levels of microendemism and beta-diversity in Andean amphibians (Fig. 8), and the region likely has many species still to be discovered (Rodríguez & Catenazzi, 2017). The addition of the two new species described herein also increases the number of species of Phrynopus known from Cordillera Yanachaga to seven (Phrynopus auriculatus, P. badius, P. bracki, P. manuelriosi sp. nov., P. melanoinguinis sp. nov., P. miroslawae, and P. tribulosus). This region shows the highest regional species diversity of Phrynopus, along with Cordillera de Carpish (P. daemon, P. dagmarae, P. interstinctus, P. kauneorum, P. lapidoides, P. unchog, and P. vestigiatus), and followed by Río Abiseo National Park (P. anancites, P.capitalis, P. dumicola, P. personatus, P. valquii). Moreover, Santa Bárbara is the only case of a type locality shared by three species of Phrynopus (P. manuelriosi sp. nov., P. melanoinguinis sp. nov., and P. miroslawae). The sympatric P. auriculatus and P. bracki occur 38 km airline SE of Santa Bárbara, 5.5 km E Oxapampa 2,600 m, on mountains at the opposite side of the Oxapampa valley (Chaparro, Padial & De la Riva, 2008), while P. badius is closer at the eastern margin of Río Huancabamba (20 km airline SE of Santa Bárbara). Santa Bárbara is located in the northwestern extreme of the Peruvian Yanachaga-Chemillén National Park (see Fig. 5) and, like in other regions throughout the Peruvian Andes, montane habitats are continuously destructed due to the increase of agricultural land and cattle ranching (Dillon et al., 1995; Venegas, 2007; Weigend, Rodríguez & Arana, 2005), which constitutes a serious threat to the species occurring therein. During our survey of amphibians and reptiles for four days in Santa Bárbara, we recorded several forest fires in the buffer zone of Yanachaga-Chemillén National Park, and we also found remains of forest fire in the grasslands and at the tree line within the national park limits (an area deemed banned for resource use). Although both species are present within the park, evidence of habitat destruction in the area raises concerns about whether these species are truly protected. Many Peruvian species of high Andean strabomantid frogs in the genera Bryophryne, Qosqophryne, Psychrophrynella, Phrynopus, and some Pristimantis have highly restricted ranges; hence, it is unlikely to find the new species of Phrynopus described herein in other parts of the national park or its surroundings. Although we agree that it is important to keep records of species that are included in threatened categories lists (e.g., International Union for Conservation of Nature (IUCN) Red List, Peruvian government threatened species list) or in natural protected areas (Aguilar et al., 2010; Von May et al. 2008), we believe that the fact that these species are legally protected will not grant their survival, especially if there are no means to secure the park boundaries from intrusions. This needs to be considered when classifying these frogs in threat categories or evaluations for becoming a priority for protection in governmental conservation plans.

Conclusions

We describe two new species of the Andean genus Phrynopus, P. manuelriosi sp. nov. and P. melanoinguinis sp. nov., based on robust morphological and molecular evidence. The new species occur sympatrically in the Andean elfin forest at an elevation of 3,280 m, in the Yanachaga-Chemillen National Park in Departamento de Pasco, Peru. Two other species of the genus, P. miroslawae and P. tribulosus, are also known to occur in the same locality. Phrynopus melanoinguinis, P. miroslawae and P. tribulosus are terrestrial, as is typical for most members of the genus, whereas P. manuelriosi possesses arboreal habitus. Despite their discovery within the boundaries of a national park, the long-term survival of these new species is not guaranteed if the borders of the protected area are not well protected. Fires observed both within the park and in its buffer zone underscore the urgent need for effective protection measures, particularly given the restricted distribution ranges characteristic of Phrynopus species. This highlights the vulnerability of these frogs and the critical importance of conserving their fragile habitats.

Supplemental Information

Supplemental Information 1 Primers used to amplify mitochondrial and nuclear genes fragments

Supplemental Information 2 Specimens examined in this study

Supplemental Information 3 GenBank accession numbers of species sampled in this study

Supplemental Information 4 Sequences used in the phylogenetic analysis and submitted to GenBank

Supplemental Information 5 Partitions

Supplemental Information 6 Alignment

Supplemental Information 7 Script

For allowing access to herpetological collections, we are grateful to J. Córdova and C. Aguilar (MUSM). We thank K. Siu-Ting and J.M. Padial for helpful comments on an earlier version of the manuscript. We are grateful to the staff of the Servicio Nacional de Áreas Naturales Protegidas por el Estado (SERNANP), especially the rangers and volunteers for their cooperation and for the research permits. We also thank to the staff of Consultores Asociados en Naturaleza y Desarrollo (CANDES) and Willy Nañez of CORBIDI for the logistic support in the field, and Daniel Matos and Abel Orihuela for the coordination with the SERNANP.

Additional Information and Declarations

Competing Interests

Author Contributions

Animal Ethics

Field Study Permissions

DNA Deposition

Data Availability

New Species Registration

The authors declare there are no competing interests.

Pablo Venegas conceived and designed the experiments, performed the experiments, analyzed the data, prepared figures and/or tables, authored or reviewed drafts of the article, and approved the final draft.

Luis Alberto García Ayachi conceived and designed the experiments, analyzed the data, prepared figures and/or tables, authored or reviewed drafts of the article, and approved the final draft.

Lesly Lujan conceived and designed the experiments, performed the experiments, analyzed the data, authored or reviewed drafts of the article, and approved the final draft.

Vilma Duran conceived and designed the experiments, performed the experiments, analyzed the data, authored or reviewed drafts of the article, and approved the final draft.

Ana Motta conceived and designed the experiments, analyzed the data, prepared figures and/or tables, authored or reviewed drafts of the article, and approved the final draft.

The following information was supplied relating to ethical approvals (i.e., approving body and any reference numbers):

Our research was approved by the Institutional Animal Care and Use Committee of University of Kansas.

AUS 279-01.

The following information was supplied relating to field study approvals (i.e., approving body and any reference numbers):

We obtained our research permit through the Dirección General Forestal y de Fauna Silvestre, Ministerio de Agricultura y Riego, Peru, which issued the Contrato de Acceso Marco a Recursos Genéticos. Approval number: 359-2013-MINAGRI-DGFFS-DGEFFS.

The following information was supplied regarding the deposition of DNA sequences:

The sequences are available at GenBank: PV034806, PV036306, PV230486, PV230487, PV034827, PV034805, PV036305, PV230484, PV230485, PV034826, PV036304, PV230482, PV230483, PV034825, Phrynopus sp. ma AM-2025, Phrynopus sp. me AM-2025.

The following information was supplied regarding data availability:

The alignments, script, and output files for the phylogenetic analysis, including partitions and trees are available in the Supplemental Files.

The following information was supplied regarding the registration of a newly described species:

Publication LSID: urn:lsid:zoobank.org:pub:513EDEF4-3F07-4097-8902-8DFED468E4C3,

Phrynopus manuelriosi LSID: urn:lsid:zoobank.org:act:86453AE1-9A6A-4ABD-9681-CF9E372B877D

Phrynopus melanoinguinis LSID: urn:lsid:zoobank.org:act:13A87303-F889-41BD-A67E-F00D4C611F17.

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
