# Peer review of "Two new sympatric species of Phrynopus (Anura: Strabomantidae) from the Elfin Forests of Cordillera de Yanachaga in central Peru"

_PeerJ, doi:10.7717/peerj.20250_

## Round 0.1 · original submission · Minor Revisions

· Academic Editor

Minor Revisions

I have read the manuscript and all three reviews and agree that this manuscript only needs a minor revision. However, please pay close attention to all comments made by the three reviewers, which will result in an improved paper.

·

Basic reporting

A final check of grammar should fix some remaining minor issues.

I suggest to improve Figure 1 (phylogenetic tree) by adding voucher numbers to all included terminals of Phrynopus at least. In some cases these are lacking. To provide respective reference, I also suggest to modfify Appendix 2 by adding voucher numbers for used sequences.

Experimental design

no comment

Validity of the findings

no comment

Additional comments

This a well-written manuscript, containing all the relevant literature. The structure and use of the English language is good. Figures are of high quality and adequately illustrate the findings. The manuscript provides substantial new scientific insight and all methods applied are more than adequate to justify the conclusions.
I provided some comments, corrections and suggestions in the attached manuscript file. These are all of minor concern and should be easy to address by the authors. I recommend to the authors to use the term 'lineage divergence' and to consider including uncorrected p-distances of the 16S barcoding gene.
In summary, this is a very nice piece of research and I am looking forward to see it published.

·

Basic reporting

This manuscript describes two new species of Andean strabomantid frogs from Peru in the genus Phrynopus, contributing to the knowledge of this speciose group of anurans. Although one of the new species is represented by a single individual, the authors make a good justification to describe it as new, based on genetic and morphological evidence. The text is clear and well written (I made minor corrections in the Word document, to improve clarity throughout the MS), the conclusions derive from the data and analyses performed, which are pertinent. The figures are of good quality and useful.
In conclusion, I recommend publication after minor revision.

Experimental design

This is not an experimental work.

Validity of the findings

No comment.

Additional comments

I recommend authors to read carefully the comments and corrections provided in order to improve the manuscript.

·

Basic reporting

While the Introduction is informative and provides a solid taxonomic and biogeographic overview of Phrynopus, it lacks a clear and explicit identification of the knowledge gap the study aims to fill. The justification for the phylogenetic and morphological investigation could be strengthened by more explicitly framing what is unknown about the genus’ diversity in the Cordillera Yanachaga, especially in comparison to other Andean regions.
Suggestion: Expand lines 68–79 to clearly identify the gap (e.g., unexamined regions, previous misidentifications, incomplete phylogenies) and how the current study addresses it.

The manuscript is generally well written, but several sections include overly long or complex sentences that could be simplified for clarity and accessibility to an international audience.
For example:
Line 23: “heel bearing one or two subconical tubercles…” could be rephrased to “the heel bears one or two subconical tubercles…”

Line 128: “and the two species we name herein” would be clearer as “including the two new species described in this study.”
I recommend a thorough language revision, either by a native speaker familiar with taxonomic herpetology or via a professional editing service.

The abstract includes some background better suited for the Introduction. Also, parts of the Results section (e.g., taxonomic diagnoses) overlap heavily with the Discussion; consider reducing redundancy by focusing Discussion on broader implications (e.g., microendemism, conservation).

Try Summarizing key differences in diagnoses and move comparisons with sympatric species primarily to the Discussion.

Experimental design

No comment

Validity of the findings

The study is a well-executed example of integrative taxonomy, using both molecular and morphological data to describe new species.

The sampling effort and fieldwork in a remote and biodiverse area are commendable.

The phylogenetic analysis is methodologically sound, and species diagnoses are detailed and comparative.

The manuscript contributes valuable knowledge to Andean herpetology and highlights important conservation issues in montane Peru.

Additional comments

The manuscript presents important new findings in Andean amphibian diversity. However, the clarity of the research rationale, and linguistic expression,should be improved.

---

## Round 0.2 · accepted · Accept

· Academic Editor

Accept

Dear Dr. Motta,

Thank you, that you have addressed all of the reviewers' comments and we happy with the current version of your manuscript, which is ready for publication.

Best regards,
Alexander Ereskovsky

·

Basic reporting

Everything fine. The authors did a good job in revising several issues that were of minor concern anyway.

Experimental design

Everything fine.

Validity of the findings

Everything fine.

Additional comments

I had a quick look at the revised version of this manuscript. In summary, the authors did a good job in revising the several minor issues. However, I feel that in the response letter, the authors confused evolutionary lineage divergence with genetic distance in the 16S barcoding gene. This is, however, of minor concern in the manuscript itself and does not need any additional fix.

Concerning the ICZN definitions of Definition and Diagnosis, these can be found in the glossary of the ICZN and are:

Definition, n.: A statement in words that purports to give those characters which, in combination, uniquely distinguish a taxon [Arts. 12, 13].

Diagnosis, n.: A statement in words that purports to give those characters that differentiate the taxon from other taxa with which it is likely to be confused.

Given these definitions, in this article, the "Diagnosis" section should be called "Definition", and the "Comparisons" section should be called "Diagnosis". However, leaving the terms as they are now in the manuscript will not cause any damage.

The authors may consider referring to the family Craugastoridae for the genus Phrynopus. The phylogenetic relationships are somewhat ambiguous, but for now, it seems more reasonable to use Craugastoridae instead of Strabomantidae.

Best regards,
Jörn.